# PersonaMath: Enhancing Math Reasoning through Persona-Driven Data Augmentation

## Abstract

While closed-source Large Language Models (LLMs) demonstrate strong mathematical problem-solving abilities, open-source models continue to struggle with such tasks. To bridge this gap, we propose a data augmentation approach and introduce PersonaMathQA, a dataset derived from MATH and GSM8K, on which we train the PersonaMath models. Our approach consists of two stages: the first stage is learning from Persona Diversification, and the second stage is learning from Reflection. In the first stage, we regenerate detailed chain-of-thought (CoT) solutions as instructions using a closed-source LLM and introduce a novel persona-driven data augmentation technique to enhance the dataset's quantity and diversity. In the second stage, we incorporate reflection to fully leverage more challenging and valuable questions. Evaluation of our PersonaMath models on MATH and GSM8K reveals that the PersonaMath-7B model (based on LLaMA-2-7B) achieves an accuracy of 24.2% on MATH and 68.7% on GSM8K, surpassing all baseline methods and achieving state-of-the-art performance. Notably, our dataset contains only 70.3K data points—merely 17.8% of MetaMathQA and 27% of MathInstruct—yet our model outperforms these baselines, demonstrating the high quality and diversity of our dataset, which enables more efficient model training. We open-source the PersonaMathQA dataset, PersonaMath models, and our code for public usage.

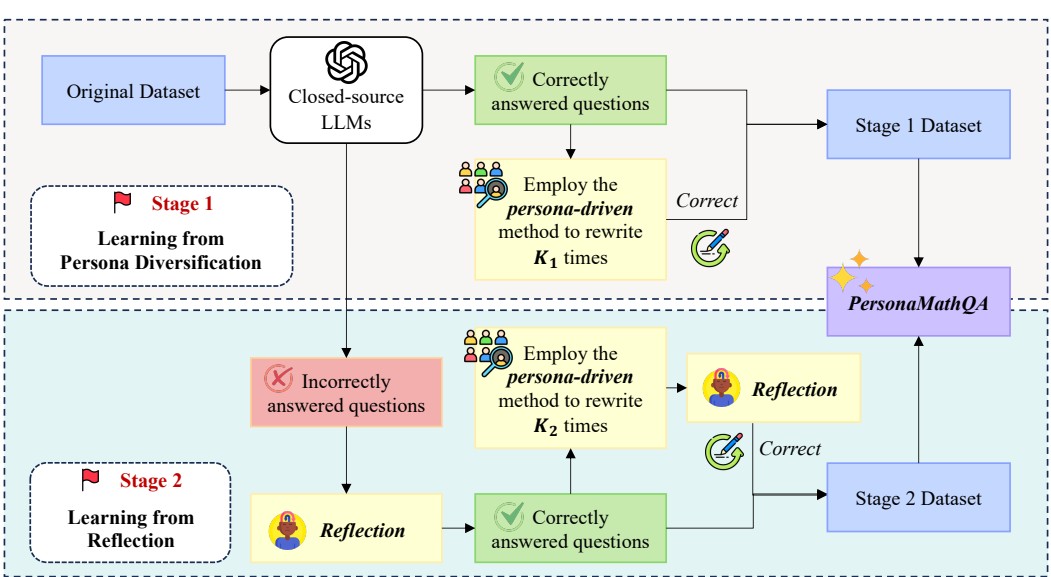

Figure 1: The framework of our data augmentation method. The method consists of two stages: Stage 1 (top) and Stage 2 (bottom). Stage 1 focuses on using closed-source LLMs to automatically generate detailed CoT solutions and apply our persona-driven rewriting method to rephrase the questions. Stage 2 focuses on reflection. The data from both stages are then combined to form our PersonaMathQA dataset.

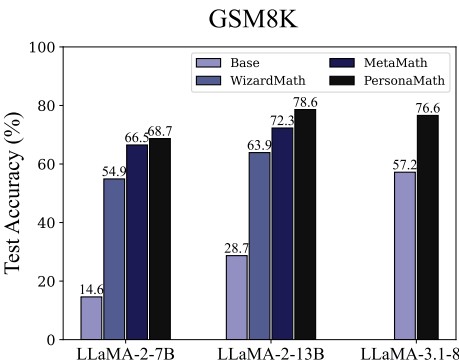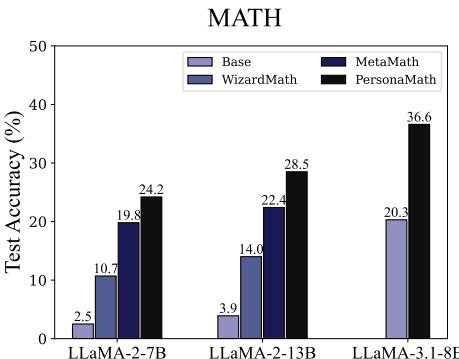

Figure 2: The superior performance of our PersonaMath models in comparison to other models. Among all models of the same size, our model achieves the highest test accuracy, demonstrating state-of-the-art performance.

# 1 INTRODUCTION

*"There are a thousand Hamlets in a thousand people's eyes"*

*Shakespeare*

Large language models (LLMs) have rapidly advanced in recent years, showing impressive abilities across various NLP tasks, including open domain conversation (Brown et al., 2020; Ouyang et al., 2022), coding (Chen et al., 2021; Rozière et al., 2024; Li et al., 2023a) and math (Luo et al., 2023; Yu et al., 2024; Imani et al., 2023). Among these tasks, solving math problems stands out as particularly challenging due to its complexity and the requirement for multi-step reasoning to reach a solution. While some closed-source models, such as GPT-4o (OpenAI, 2024a), Claude 3.5 Sonnet (Anthropic, 2024), and Gemini 1.5 Pro (Reid et al., 2024), have demonstrated strong math-solving capabilities, current open-source models (e.g., LLaMA (Touvron et al., 2023; Dubey et al., 2024)) continue to struggle in this area. Therefore, enhancing the math problem-solving abilities of open-source models is a prominent desiderata.

A widely adopted and effective approach for improving the math-solving capabilities of open-source models is fine-tuning, owing to the accessibility of their weights (Yuan et al., 2023; Yue et al., 2023; Luo et al., 2023; Yu et al., 2024). However, directly fine-tuning on the original dataset leads to only minimal improvements (Touvron et al., 2023), as solving complex math problems requires multiple reasoning steps, which the original dataset does not adequately capture. Furthermore, the limited number of samples in the original dataset is often insufficient for models to fully grasp how to solve mathematical problems. To address this, recent studies have demonstrated promising results by bootstrapping the original dataset (Yu et al., 2024), generating more detailed Chain of Thought (CoT) solutions (Wei et al., 2023), and training models on the enhanced dataset. However, directly bootstrapping from the original problems can reduce the diversity of the enhanced dataset, necessitating a large amount of data to achieve effective training, which is resource-intensive and inefficient.

In this paper, we propose a data augmentation method designed to enable models to learn more efficiently and effectively from enhanced datasets. The key to our approach lies in improving both the quality and diversity of the data. To achieve this, we introduce a persona-driven approach and create PersonaMathQA, a dataset characterized by high efficiency and diversity. Recently, Role-Playing LLMs that incorporate personas have gained significant attention, as leveraging personas allows LLMs to adapt to persona-specific scenarios and generate unique, diverse answers from various perspectives. Moreover, recent research has further highlighted the potential of Role-Playing LLMs. From extensive web data, Chan et al. (2024) automatically created Persona Hub, a collection of 1 billion diverse personas, with 200,000 personas released publicly. Building on this vast collection, our proposed data augmentation method can generate data from a wide range of diverse perspectives, encompassing various ethnicities, occupations, age groups, and more.

Specifically, our method is divided into two stages. In the first stage, we focus on empowering open-source models to **Learn from Persona Diversification** by proposing a persona-driven method. We

start by using a powerful closed-source LLM to generate detailed CoT solutions for the math problems within the original dataset. After identifying the correctly answered questions, we apply the persona-driven method to prompt the closed-source LLM to rewrite these questions using specific and diverse personas. We then regenerate the CoT solutions and incorporate both the correctly answered questions and their original questions in our dataset. In the second stage, we shift our focus to enabling open-source models to **Learn from Reflection**. For questions that the closed-source LLM answered incorrectly in the first stage, we allow the LLM to reflect on its mistakes and attempt to regenerate the correct answers. For those questions answered correctly post-reflection, which are inherently more challenging than those answered correctly in the first stage, we aim to increase their weight in the final dataset. This strategy facilitates more effective learning of these difficult questions by open-source LLMs through training. To achieve this, we increase the number of rewrites for these challenging questions and follow similar procedures as in the first stage, where regeneration is accomplished through reflection.

Our main contributions are summarized as follows:

- We propose a novel persona-driven data augmentation method, resulting in the creation of a high-quality and efficient dataset, PersonaMathQA.
- Using our PersonaMathQA dataset, we trained LLaMA and Qwen2.5, resulting in the PersonaMath models. We evaluated the math problem-solving capabilities of the PersonaMath models on two widely recognized math benchmarks, MATH (Hendrycks et al., 2021) and GSM8K (Cobbe et al., 2021). Despite our dataset being only 17.79% the size of previous state-of-the-art datasets, PersonaMath outperformed them on both benchmarks, setting a new SOTA.
- We demonstrated the effectiveness of the persona-driven method in data rewriting. Our approach not only improves dataset diversity and quality but also reduces training costs compared to other rewriting methods. Our work highlights the significant potential of persona-driven methods and offers valuable insights for future research.

## 2 RELATED WORK

**Large Language Models for Mathematical Reasoning.** Among various NLP tasks, solving math problems has consistently been regarded as one of the most challenging. Current cutting-edge closed-source LLMs, such as o1 (OpenAI, 2024c), GPT-4o (OpenAI, 2024a), Claude 3.5 Sonnet (Anthropic, 2024), and Gemini 1.5 Pro (Reid et al., 2024), exhibit strong math problem-solving capabilities. Nonetheless, there remains substantial work to further enhance these capabilities through various strategies, including preprocessing math questions (An et al., 2023), utilizing more advanced prompts (Ling et al., 2017; Yao et al., 2023), employing external tools (Yamauchi et al., 2023; He-Yueya et al., 2023; Chen et al., 2022), and enhancing overall interactions (Wu et al., 2024). In contrast, open-source models with fewer parameters still struggle with solving math problems. Consequently, numerous studies, including our own, aim to bridge this gap, enabling open-source models to achieve robust math problem-solving capabilities after fine-tuning. To enhance the effectiveness of fine-tuning, past approaches have primarily focused on generating intermediate steps for answers (Nye et al., 2021; Zhang et al., 2023; Yang et al., 2023b; Lewkowycz et al., 2022), fine-tuning across multiple datasets (Mishra et al., 2022; Yue et al., 2023), employing teacher-student knowledge distillation (Imani et al., 2023), and learning from enhanced datasets (Imani et al., 2023; Raiyan et al., 2023; Yu et al., 2024), as explored in our paper.

**Role-Playing Large Language Models.** Recently, Role-Playing LLMs that integrate personas have gained significant attention. Specifically, Role-Playing LLMs involve directly incorporating personas into the prompts of LLMs, enabling them to generate unique, role-specific content within a designated context. This method is straightforward to implement, requiring only prompt modifications, yet it yields impressive results. For instance, Dong et al. (2024) employed multiple LLM agents, assigning each to play a specialized expert role, allowing them to collaboratively tackle complex code generation tasks. Similarly, ChatDev (Qian et al., 2024) segmented the software development process into four stages—designing, coding, testing, and documenting—and proposed a Chat Chain to facilitate communication among agents responsible for each stage, ultimately enabling cooperative program development. Role-Playing LLMs are also frequently utilized in gaming contexts. For instance, Wang et al. (2023) has an LLM assume the role of a general assistant, continuously exploring the Minecraft game world to acquire skills and survive longer. In the medical domain, Tang et al. (2024) simulates real-life scenarios by dividing the diagnostic reasoning process

into five stages: expert gathering, analysis proposition, report summarization, collaborative consultation, and decision-making. Multiple LLM agents are then assigned different roles to collaborate on the diagnostic process.

# 3 METHOD

In this section, we provide a detailed description of how we built PersonaMathQA. An overview of our approach is illustrated in Figure 1. Our method, which enhances data using a persona-driven approach, is divided into two stages. Using this approach, we constructed the PersonaMathQA dataset based on the MATH and GSM8K datasets, and fine-tuned the PersonaMath models on it.

## 3.1 STAGE 1: LEARNING FROM PERSONA DIVERSIFICATION

**Limitations of the Original Dataset.** Although fine-tuning open-source models can significantly enhance their performance across various tasks, improving their math problem-solving capabilities through fine-tuning alone is particularly challenging. One reason is that ***math problems are inherently complex*** and often require intricate multi-step reasoning to arrive at the correct answer (Ahn et al., 2024), ***with errors at any step potentially leading to incorrect results.*** Another significant challenge is the limitation of current training datasets. Effective math problem-solving requires detailed step-by-step reasoning, yet ***existing datasets often lack such comprehensive solutions, often providing only final answers or insufficiently detailed manual solutions.*** Consequently, models trained on such datasets struggle to develop step-by-step reasoning skills, yielding minimal improvements from fine-tuning alone (Touvron et al., 2023). Manually labeling detailed step-by-step solutions is time-consuming and labor-intensive. To address this, we use a closed-source LLM with advanced mathematical problem-solving abilities to automatically generate detailed CoT solutions, thereby enhancing the training dataset with comprehensive, step-by-step reasoning. The prompt we used is shown below.

---

**Prompt for Inference**

*Please provide a detailed, step-by-step explanation for the following math problem. At the end of the explanation, present the final answer enclosed in \\boxed{} \n Math problem:*

---

**Persona-Driven Data Augmentation Method.** After obtaining the CoT solutions automatically generated by the LLM, we first filter out the correct answers and add them to our PersonaMathQA dataset. However, this initial training data is insufficient on its own. To create high-quality enhanced datasets and avoid excessive time and resource expenditure, we employ a data augmentation method in which the LLM automatically rewrites questions to generate new ones. Previous studies have shown that directly rewriting questions is inefficient (Yu et al., 2024), as it requires a large volume of data to significantly boost model performance. This inefficiency arises from the homogeneous nature of rewritten questions, which often lack diversity. To address this issue and augment the dataset more effectively, we propose a novel persona-driven method. Inspired by the observation that introducing persona-related scenarios to the LLM can prompt it to generate unique and persona-specific content, we theorize that with a sufficient number of diverse personas, we can generate a vast amount of varied content. A recent study (Chan et al., 2024) that released 200,000 diverse personas created from extensive web data provides a solid foundation for our approach. These personas encompass various nationalities, races, religions, occupations, age groups, and more. Leveraging this diversity, our method can create varied questions, yielding significant improvements with less data, demonstrating the principle of "less is more". Additionally, our persona-driven method is straightforward to implement, allowing others to create their own augmented datasets using our approach. The prompt we used is shown below.

---

**Prompt for Rewriting**

*Math problem: {problem} \nPlease rephrase the above math problem with the following persona:\n{persona}*

---

|  | Correct | Incorrect |
|---|---|---|
| Average Level | 3.22 | 4.28 |

Table 1: The average difficulty levels of correctly and incorrectly answered questions in the MATH dataset during the reasoning stage of Stage 1. "Correct" refers to the questions answered correctly, while "Incorrect" refers to those answered incorrectly.

We instruct the LLM to rewrite each question $K_1$ times, introducing different personas each time to maximize the diversity of the rewritten questions. Subsequently, the LLM reasons through these rewritten questions and generates detailed CoT solutions. We select the correct solutions along with their corresponding questions to include in our PersonaMathQA dataset. Questions that are answered incorrectly after rewriting are discarded.

## 3.2 STAGE 2: LEARNING FROM REFLECTION

**Shifting Focus to Incorrectly Answered Questions.** In Stage 1, we first have the LLM reason through the original dataset and identify questions with correct answers. Questions answered incorrectly are set aside for further analysis. For a detailed analysis, we use the MATH dataset, where each problem is classified into difficulty levels ranging from "1" to "5". We calculated the average difficulty levels of correctly and incorrectly answered questions during the reasoning stage. The results, shown in Table 1, indicate that the average difficulty level of correctly answered questions is 3.22, whereas incorrectly answered questions have an average level of 4.28. This suggests that the incorrectly answered questions are inherently more challenging, making them harder for the LLM to solve. However, this also makes them more valuable, as learning to solve these complex problems can significantly enhance the model's step-by-step reasoning ability.

**Reflecting on Errors and Regenerating Solutions.** To explore and utilize the potential value in these incorrectly answered questions, we prompt the LLM to reflect on its mistakes and attempt to provide correct answers. This approach is inspired by previous research demonstrating that LLMs can self-reflect and self-correct (Shinn et al., 2023; Li et al., 2023b). Specifically, for the questions that the LLM answered incorrectly in Stage 1, we return the wrong solution to the LLM, inform it that the answer is incorrect, and prompt it to reflect on its reasoning process and regenerate a detailed CoT solution. The prompt we used is shown below, where the explanation is replaced with the incorrect solution.

> **Prompt for Reflection**
>
> *The following input consists of a math problem and a corresponding explanation. However, this explanation is incorrect, please reflect on its errors and then generate a corrected, detailed, step-by-step explanation for the following math problem. Divide your response into two parts: Review of Incorrect Explanation and Corrected Explanation. At the end of the explanation, present the final answer enclosed in \\boxed{}.\nMath Problem: {problem}\nIncorrect Explanation: {explanation}*

In this step, the LLM's response is divided into two parts: "Review of Incorrect Explanation" and "Corrected Explanation". We use only the "Corrected Explanation" as the regenerated solution from the LLM. However, it is important to note that while the "Review of Incorrect Explanation" is not used directly, it is an indispensable part of the reflection process. This part allows the LLM to summarize its mistakes, which is crucial for generating the subsequent "Corrected Explanation". Correctly answered questions are then added to our PersonaMathQA dataset, while questions that remain unanswered correctly by the LLM are discarded. It's worth noting that previous studies have also explored generating the correct solution process by directly providing the LLM with the correct answer (Zhang et al., 2024). However, this method carries risks due to LLM hallucinations, where the LLM may produce an incorrect solution process while providing the correct final answer. Such data can be misleading and detrimental. Therefore, we discard questions that cannot be answered correctly. For the questions answered correctly in Stage 2, we rewrite them as we did in Stage 1. The key differences is that in Stage 2, we rewrite the questions $K_2$ times, where $K_2$ is greater than $K_1$,

| Dataset | Stage 1 | | Stage 2 | | Overall |
| --- | --- | --- | --- | --- | --- |
| | Inference | Rewrite | Reflection | Rewrite | |
| PersonaMathQA-GSM8K | 6.6K | 30.8K | 0.1K | 1.1K | 38.7K |
| PersonaMathQA-MATH | 5.4K | 23.8K | 0.2K | 1.9K | 31.8K |
| PersonaMathQA | 12.1K | 54.6K | 0.3K | 3.1K | 70.3K |

Table 2: The detailed composition of each component of the PersonaMathQA dataset. This table shows the number of questions with correct answers that were added to PersonaMathQA during the final step of each phase.

the number of rewrites in Stage 1. As discussed earlier, this is because Stage 2 problems are more difficult, and the performance gains from training on these more challenging problems are higher. Therefore, increasing the number of rewrites in Stage 2 helps to more effectively enhance model performance after training. We then implement our reflection framework to regenerate the solutions and select those with the correct answers. For computational efficiency, we use the incorrect solution from the original question as the incorrect solution for the rewritten questions.

## 4 EXPERIMENTS

### 4.1 TRAINING AND EVALUATION SETUP

We fine-tune open-source models using the PersonaMathQA dataset to develop the PersonaMath models. We utilize the training prompt provided in (Taori et al., 2023) and fine-tune the model by maximizing the log-likelihood of the reasoning path given the question, expressed as $\mathcal{L}(\theta) = \Sigma_{(q,r)\in\text{PersonaMathQA}} \log P(r|q;\theta)$. Here, $\theta$ represents the parameters of the open-source model, $q$ represents the question, and $r$ denotes the correct solution generated by the closed-source LLM. This approach ensures our method is readily adaptable for training any open-source model. The training prompt is provided below, where the "*instruction*" is replaced by questions from the PersonaMathQA dataset, and the corresponding solutions follow after "*Response:* ".

> ### Training Prompt
>
> *Below is an instruction that describes a task. Write a response that appropriately completes the request.\n\n### Instruction:\n{instruction}\n\n### Response:*

The CoT prompt used for evaluating the trained model is also sourced from (Taori et al., 2023), as shown below. The "*instruction*" is replaced by questions from the test set.

> ### Evaluation Prompt
>
> *Below is an instruction that describes a task. Write a response that appropriately completes the request.\n\n### Instruction:\n{instruction}\n\n### Response: Let's think step by step.*

### 4.2 EXPERIMENTAL SETUP

**Datasets.** We apply our data-enhancement method to two well-known math problem datasets, MATH (Hendrycks et al., 2021) and GSM8K (Cobbe et al., 2021), to create our PersonaMathQA dataset. GSM8K consists of problems designed by human writers, requiring between 2 and 8 steps to solve. Solutions involve a sequence of basic arithmetic operations (+, -, *, /) to arrive at the final answer. The GSM8K dataset includes approximately 7,500 training problems and about 1,000 test problems. In contrast, MATH is a more challenging dataset composed of problems from mathematics competitions, such as AMC 10, AMC 12, and AIME, covering a wide range of subjects and difficulty levels. The problems span seven subjects: Prealgebra, Algebra, Number Theory, Counting and Probability, Geometry, Intermediate Algebra, and Precalculus, and are classified into five diffi-

culty levels, with higher numbers indicating greater complexity. MATH consists of 7,500 training problems and 5,000 test problems.

**Models.** For inference, rewriting, and reflection, we utilized the closed-source LLM GPT-4o-mini-2024-07-18 (OpenAI, 2024b) with a temperature setting of 0.7. To evaluate our approach, we fine-tuned the open-source models LLaMA-2-7B and LLaMA-2-13B (Touvron et al., 2023) on our dataset. Additionally, to facilitate future comparisons, we also fine-tuned the current state-of-the-art open-source model LLaMA-3.1-8B (Dubey et al., 2024) and Qwen2.5-7B (Team, 2024). We employed DeepSpeed for training, applying the ZeRO-2 stage for LLaMA-2-7B, LLaMA-3.1-8B and Qwen2.5-7B (Team, 2024), and the ZeRO-3 stage for LLaMA-2-13B to optimize cost and efficiency (Rajbhandari et al., 2020). All models were trained using four A800 80GB PCIe GPUs. Further details are provided in Appendix A.

**Baselines.** To evaluate our proposed method comprehensively, we establish a baseline that includes a range of closed-source models, open-source models, and state-of-the-art methods for training open-source models to solve mathematical problems. The closed-source models include o1-preview (OpenAI, 2024c), GPT-4o (OpenAI, 2024a), PaLM-2 (Anil et al., 2023), Claude 3.5 Sonnet (Anthropic, 2024), and others. The open-source models encompass LLaMA-2 (Touvron et al., 2023), LLaMA-3.1 (Dubey et al., 2024), Qwen2.5 (Team, 2024), Mistral Large 2 (AI, 2024), and others. We also compared our method against three prominent techniques:

- **WizardMath** (Luo et al., 2023): This approach enhances the mathematical reasoning capabilities of LLaMA-2 using a Reinforcement Learning from Evol-Instruct Feedback (RLEIF) method. RLEIF involves three steps: (1) supervised fine-tuning (SFT), (2) training an Instruction Reward Model (IRM) and a Process-supervised Reward Model (PRM), and (3) active Evol-Instruct and reinforcement learning via proximal policy optimization (PPO).
- **MAmmoTH** (Yue et al., 2023): The MAmmoTH models are trained on MathInstruct, a dataset compiled from 13 math datasets that include intermediate rationales. MAmmoTH uses a combination of chain-of-thought (CoT) and program-of-thought (PoT) rationales. While they evaluate models using both approaches, we focus on comparing results where CoT is used for a fairer comparison.
- **MetaMath** (Yu et al., 2024): This approach involves bootstrapping mathematical questions by rewriting them from multiple perspectives, creating a new dataset called MetaMathQA. The LLaMA-2 models are then fine-tuned on MetaMathQA to produce the MetaMath models.

Additionally, we compared our method with several basic fine-tuning approaches:

- **Supervised Fine-Tuning (SFT)**: This method involves fine-tuning the models using the training sets from the original GSM8K or MATH datasets.
- **Rejection Sampling Fine-Tuning (RFT)** (Yuan et al., 2023): This approach generates and collects correct reasoning paths as augmented data for fine-tuning.

### 4.3 MAIN RESULTS

Our PersonaMathQA dataset consists of 70.3K samples, with its detailed composition shown in Table 2. Table 3 presents the test accuracy of our method compared to a range of baselines on the GSM8K and MATH datasets. The results demonstrate that, compared to the pre-trained models, our trained model's mathematical problem-solving abilities have significantly improved, surpassing most open-source models of the same size. Additionally, our method outperforms the three baseline approaches, achieving state-of-the-art results. Notably, in the two baselines that also employ data enhancement techniques, the MathInstruct dataset used in MAmmoTH (Yue et al., 2023) contains 260K samples, and the MetaMathQA (Yu et al., 2024) dataset includes 395K samples. Despite our PersonaMathQA dataset having only 70.3K samples—27.0% of MathInstruct and 17.8% of MetaMathQA—our approach yields superior results. This indicates that our dataset is of higher quality, enabling the model to acquire more robust mathematical problem-solving skills with fewer samples. Furthermore, it suggests that expanding the PersonaMathQA dataset using our method could lead to even greater improvements in model performance.

### 4.4 DISCUSSION ON DATASET DIVERSITY

One notable aspect of our main results is that our method not only surpasses the baseline method MetaMath but also achieves this with a significantly smaller dataset, containing only 17.8% of their

| Model | Params | Base | GSM8K | MATH |
|---|---|---|---|---|
| *closed-source models* | | | | |
| GPT-4 (OpenAI et al., 2024) | - | - | 92.0 | 42.5 |
| GPT-4o (OpenAI, 2024a) | - | - | - | 76.6 |
| GPT-4o mini (OpenAI, 2024b) | - | - | - | 70.2 |
| o1 (OpenAI, 2024c) | - | - | - | 94.8 |
| PaLM-2 (Anil et al., 2023) | 540B | - | 80.7 | 34.3 |
| Claude 3.5 Sonnet (Anthropic, 2024) | - | - | 96.4 | 71.1 |
| Gemini 1.5 Pro (Reid et al., 2024) | - | - | 90.8 | 67.7 |
| *open-source models (6-9B)* | | | | |
| LLaMA-2-7B (Touvron et al., 2023) | 7B | - | 14.6 | 2.5 |
| LLaMA-3.1-8B (Dubey et al., 2024) | 8B | - | 57.2 | 20.3 |
| Code-LLaMA (Rozière et al., 2024) | 7B | - | 25.2 | 13.0 |
| GLM-4-9B (GLM et al., 2024) | 9B | - | 84.0 | 30.4 |
| Qwen2.5-7B (Team, 2024) | 7B | - | 85.4 | 49.8 |
| Baichuan 2 (Yang et al., 2023a) | 7B | - | 24.5 | 5.6 |
| DeepSeek-V2 (DeepSeek-AI et al., 2024) | 16B | - | 41.1 | 17.1 |
| SFT (Touvron et al., 2023) | 7B | LLaMA-2-7B | 41.6 | - |
| RFT (Yuan et al., 2023) | 7B | LLaMA-2-7B | 50.3 | - |
| WizardMath (Luo et al., 2023) | 7B | LLaMA-2-7B | 54.9 | 10.7 |
| MAmmoTH(CoT) (Yue et al., 2023) | 7B | LLaMA-2-7B | 50.5 | 10.4 |
| MetaMath (Yu et al., 2024) | 7B | LLaMA-2-7B | 66.5 | 19.8 |
| **PersonaMath** | 7B | LLaMA-2-7B | **68.7** | **24.2** |
| **PersonaMath** | 8B | LLaMA-3.1-8B | **76.6** | **36.6** |
| **PersonaMath** | 7B | Qwen2.5-7B | **84.3** | **56.6** |
| *open-source models (more than 10B)* | | | | |
| LLaMA-2-13B (Touvron et al., 2023) | 13B | - | 28.7 | 3.9 |
| LLaMA-2-70B (Touvron et al., 2023) | 70B | - | 56.8 | 13.5 |
| LLaMA-3.1-70B (Dubey et al., 2024) | 70B | - | 83.7 | 41.4 |
| Code-LLaMA (Rozière et al., 2024) | 13B | - | 36.1 | 16.4 |
| Qwen2.5-72B (Team, 2024) | 72B | - | 91.5 | 62.1 |
| Baichuan 2 (Yang et al., 2023a) | 13B | - | 52.8 | 10.1 |
| DeepSeek-V2 (DeepSeek-AI et al., 2024) | 236B | - | 79.2 | 43.6 |
| Mistral Large 2 (AI, 2024) | 123B | - | 93.0 | 71.5 |
| LLaMA-3.1-405B (Dubey et al., 2024) | 405B | - | 89.0 | 53.8 |
| SFT (Touvron et al., 2023) | 13B | LLaMA-2-13B | 50.0 | - |
| RFT (Yuan et al., 2023) | 13B | LLaMA-2-13B | 55.4 | - |
| WizardMath (Luo et al., 2023) | 13B | LLaMA-2-13B | 63.9 | 14.0 |
| MAmmoTH(CoT) (Yue et al., 2023) | 13B | LLaMA-2-13B | 56.3 | 12.9 |
| MetaMath (Yu et al., 2024) | 13B | LLaMA-2-13B | 72.3 | 22.4 |
| **PersonaMath** | 13B | LLaMA-2-13B | **78.6** | **28.5** |

Table 3: Test accuracy on GSM8K and MATH datasets. "PersonaMath" refers to our model. The primary baselines compared include WizardMath, MAmmoTH, and MetaMath. Our models demonstrate higher accuracy compared to the baselines when trained on the same underlying model

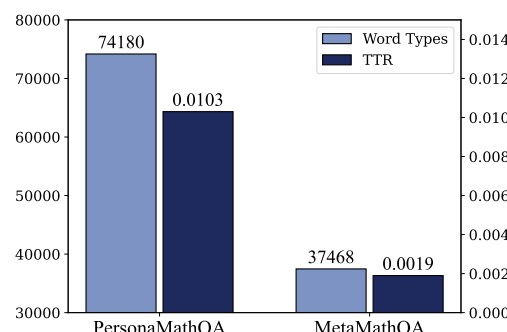
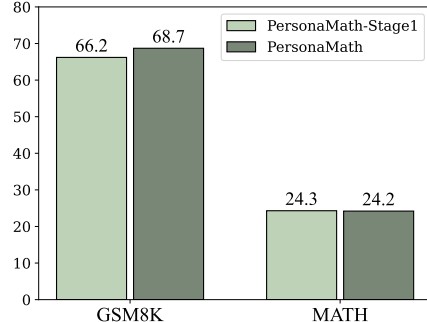

Figure 3: Comparison of Word Types and TTR between our PersonaMathQA dataset and Meta-MathQA. PersonaMathQA significantly surpasses MetaMathQA in both metrics, demonstrating its superior diversity and quality.

Figure 4: Figure 4: Results of the ablation study on the impact of Stage 2 data. Despite comprising a small portion of PersonaMathQA, Stage 2 data contributes to a 2.5% improvement in the model's test accuracy on GSM8K, underscoring its importance and necessity.

samples. To illustrate this, we conducted two experiments to assess the diversity of questions in our dataset compared to the baseline datasets. In the first experiment, we split the questions into individual words, counted the Word Types (the number of unique, non-repeating words), and calculated the Type Token Ratio (TTR), which is the ratio of Word Types to the total number of words, for both our dataset and the MetaMathQA dataset. Higher values for Word Types and TTR indicate greater diversity. Our experimental results, shown in Figure 3, reveal that the PersonaMathQA dataset significantly outperforms MetaMathQA in both Word Types and TTR.

In the second experiment, we calculated the distribution of question lengths across three datasets. The results, shown in Figure 5, where "Original" refers to the sum of MATH and GSM8K datasets, and "Normalized Frequency" refers to the frequency normalized so that the area under the histogram integrates to 1. As depicted in this figure, the question length distributions in the original dataset and MetaMathQA overlap significantly and are concentrated in the range of shorter questions, indicating a lack of uniformity in the distribution. In contrast, our dataset demonstrates a more uniform and broader distribution, featuring a higher proportion of longer questions and covering a wider variety of question lengths. Both experiments demonstrate that our dataset possesses markedly higher diversity. Consequently, our dataset is of superior quality and can achieve better results with fewer samples. The enhanced diversity of our dataset is attributed to the persona-driven method employed. As illustrated by the example below, even when starting from the same question, incorporating different personas introduces varied contexts for the LLM, leading to diverse and unique rewritten questions.

### 4.5 ABLATION STUDY: GAINS FROM STAGE 2

As discussed earlier, the data obtained in Stage 2 is highly valuable due to its increased difficulty and the fact that it includes questions that the closed-source LLM could not answer correctly in Stage 1. This challenging nature makes Stage 2 data particularly beneficial for training the model to develop robust step-by-step problem-solving abilities. Despite this, Table 2 shows that Stage 2 data constitutes only 4.8% of the total dataset, which might lead to questions about its impact. To address this, we conducted an ablation experiment using the MATH and GSM8K datasets as case studies. We applied our method to both datasets and compared the performance of two models based on LLaMA-2-7B: one trained on a dataset containing only Stage 1 data, and the other trained on a dataset containing both Stage 1 and Stage 2 data (i.e., PersonaMathQA). We then compared the performance of these models on the MATH and GSM8K datasets. The results, shown in Figure 4, indicate that while incorporating Stage 2 data does not significantly impact the model's performance on the MATH dataset, it enhances the model's test accuracy on GSM8K by 2.5%. These improvements highlight the value of incorporating Stage 2 data, which reuses incorrectly answered questions from Stage 1. Despite the relatively small size of the Stage 2 dataset, it has a substantial impact on model

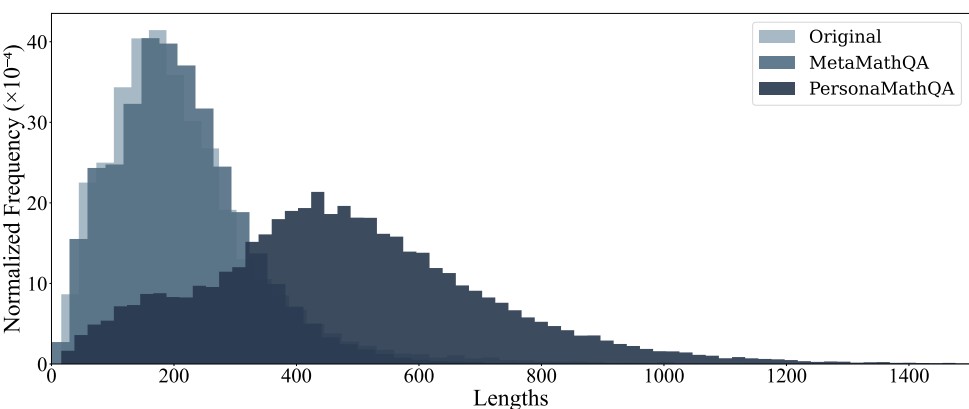

Figure 5: Comparison of the distribution of question lengths between our dataset and the two baseline datasets, where "Original" refers to the sum of the MATH and GSM8K datasets. The result shows that the distribution of question lengths in our dataset is more uniform and broader than in the two baseline datasets, indicating superior diversity.

performance. This also validates our approach of treating Stage 2 data separately and applying additional rewrites to enhance their contribution.

---

**Example of Question Rewriting**

**Original problem**: Suppose I have 6 shirts, 4 ties, and 3 pairs of pants. If an outfit requires a shirt and pants, and can either have a tie or not have a tie, how many outfits can I make?

**Direct rewrite** (example from MetaMathQA (Yu et al., 2024)): If I have 6 shirts, 4 ties, and 3 pairs of pants, and each outfit consists of a shirt and pants with the option to include a tie, how many different outfits can I create?

**Persona-driven rewrite** (Ours):

*Persona:* An author exploring dystopian futures and the ethical dilemmas presented by transhumanist technologies in their novels

*Rephrased problem:* In the narrative of a dystopian landscape where fashion serves as both a status symbol and an ethical statement, envision a character equipped with 6 distinct shirts, 4 ties that symbolize various societal allegiances, and 3 pairs of pants that reflect their personal journey. As they prepare for a pivotal encounter, each outfit must consist of a chosen shirt and pants, with the option to either adorn themselves with a tie—denoting conformity and allegiance—or to embrace a more rebellious, tie-less identity. How many unique ensembles can this character assemble, given these constraints?

---

## 5 CONCLUSION

In this paper, we advance the mathematical problem-solving capabilities of open-source LLMs by introducing a two-stage method for dataset augmentation. The first stage, Learning from Persona Diversification, involves generating detailed CoT solutions and implementing a novel persona-driven approach for rewriting questions. The second stage, Learning from Reflection, focuses on incorporating reflection to fully exploit more challenging and valuable questions. By applying this method to the MATH and GSM8K datasets, we developed PersonaMathQA, a high-quality, diverse dataset comprising only 70.3K samples. Despite its smaller size compared to baseline methods, the PersonaMath models trained on PersonaMathQA have achieved state-of-the-art performance on both datasets. Our approach demonstrates the significant potential of persona-driven methods for dataset enhancement and provides valuable insights for future research in this field.

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

# Appendix

## A  EXPERIMENTAL DETAILS

**Training Details.**   For the entire training process, we use the AdamW optimizer with 3 epochs, leveraging 4 NVIDIA A800 80GB PCIe GPUs. The learning rate is set to 2e-5 with a 3% learning rate warmup. We employed DeepSpeed for training, applying the ZeRO-2 stage for LLaMA-2-7B, LLaMA-3.1-8B, and Qwen2.5-7B, while utilizing the ZeRO-3 stage for LLaMA-2-13B to optimize cost and efficiency. For ZeRO-2, we set gradient accumulation steps to 4 with a per-device training batch size of 1, and for ZeRO-3, we set gradient accumulation steps to 16 with a per-device training batch size of 2 to accelerate the training process. The max model lengths are set to 4096 for LLaMA-2-7B and LLaMA-3.1-8B, while for LLaMA-2-13B, we set it to 2048 for training efficiency. For Qwen2.5-7B, the max model length is also set to 2048 due to its maximum supported length. For more details, please refer to our released code.

**Answer Extraction.**   It's worth noting that when we prompt the closed-source LLM to generate the CoT solutions, we always include "*present the final answer enclosed in* $\backslash\backslash boxed\{\}$" at the end of the prompt. As a result, the answer will always be enclosed in a "$\backslash\backslash$boxed$\{\}$", allowing us to simply extract the content within the "$\backslash\backslash$boxed$\{\}$" as the answer.

## B  ADDITIONAL CASE STUDIES

> **Examples of complete responses from our LLM reflection framework**
>
> **Example from MATH:**
>
> *Problem:* If $a$ and $b$ are positive integers for which $ab - 3a + 4b = 137$, what is the minimal possible value of $|a - b|$?
>
> *Incorrect Response:* We begin with the equation given in the problem: $ab - 3a + 4b = 137$. To analyze this equation, we can rearrange it to facilitate factoring: $ab - 3a + 4b - 137 = 0$. Next, we can rewrite the equation in a more manageable form, isolating $ab$: $ab = 3a - 4b + 137$. Rearranging presents: $ab - 4b = 3a + 137$, $b(a - 4) = 3a + 137$. From this, we can solve for $b$: $b = \frac{3a+137}{a-4}$. To ensure $b$ is an integer, the term $3a + 137$ must be divisible by $a - 4$. We set up the condition for divisibility. To do this, we first simplify with polynomial long division (or by applying the remainder theorem). We can divide $3a + 137$ by $a - 4$: 1. The leading term calculates as 3. 2. Multiply $3(a - 4)$: $3a - 12$. 3. Subtract from the original expression: $(3a + 137) - (3a - 12) = 137 + 12 = 149$. Thus, we can express this division as: $3a + 137 = 3(a - 4) + 149$. This indicates that the remainder, when dividing by $a - 4$, is 149. Therefore, for $b$ to be an integer, 149 must be divisible by $a - 4$. This leads us to find the positive divisors of 149. The divisors of 149 (which is a prime number) are: 1 and 149. Now we analyze each case:### Case 1: $a - 4 = 1 a = 5$. Substituting into our equation for $b$: $b = \frac{3(5)+137}{5-4} = \frac{15+137}{1} = 152$. Here, $a = 5$ and $b = 152$. We compute $|a - b|$: $|5 - 152| = 147$. ### Case 2: $a - 4 = 149 a = 153$. Substituting into the equation for $b$: $b = \frac{3(153)+137}{153-4} = \frac{459+137}{149} = \frac{596}{149} = 4$. In this case, $a = 153$ and $b = 4$. Now compute $|a - b|$: $|153 - 4| = 149$. Among the computed values of $|a - b|$, we have: - From case 1: 147. - From case 2: 149. The minimal value of $|a - b|$ is therefore: $\min(147, 149) = 147$. Finally, we write our answer in the required boxed format: $\boxed{147}$
>
> *Complete Response:*### Review of Incorrect Explanation: The incorrect explanation makes several errors in the calculations and assumptions: 1. **Divisor Calculation**: It correctly identifies that $3a + 137$ must be divisible by $a + 4$ but fails to correctly analyze how to handle the conditions when $a + 4$ equals the divisors of 125. The calculations for certain cases are also incorrect. 2. **Case Evaluations**: In Case 3 and Case 4, the evaluations of $b$ based on the respective values of $a$ are incorrect. The values of $b$ derived from these calculations do not satisfy the original equation. 3. **Final Comparison**: The final step incorrectly states the minimum without correctly evaluating all possible valid cases. ### Corrected Explanation: To solve the equation $ab - 3a + 4b = 137$ for positive integers $a$ and $b$, we will follow these steps: 1. **Rearranging the Equation**: Start with the equation: $ab - 3a + 4b = 137$. Rearranging it gives: $ab + 4b - 3a = 137$. 2. **Isolating $b$**: We can express $b$ in terms of $a$: $b(a + 4) = 3a + 137$ Thus: $b = \frac{3a+137}{a+4}$ For $b$ to be an integer, $3a + 137$ must be divisible by $a + 4$. 3. **Finding Conditions for Divisibility**: We perform polynomial long division of $3a + 137$ by $a + 4$: - The leading term is 3. - Multiply $3(a + 4) = 3a + 12$. - Subtract: $(3a + 137) - (3a + 12) = 125$ Hence: $3a + 137 = 3(a + 4) + 125$ This shows $a + 4$ must be a divisor of 125. 4. **Finding Divisors of 125**: The positive divisors of 125 are $1, 5, 25, 125$. We will set $a + 4$ equal to each divisor and solve for $a$ and $b$: - **Case 1**: $a + 4 = 1 \Rightarrow a = -3$ (not valid) - **Case 2**: $a + 4 = 5 \Rightarrow a = 1$ $b = \frac{3(1)+137}{1+4} = \frac{3+137}{5} = \frac{140}{5} = 28$ $|1 - 28| = 27$ - **Case 3**: $a + 4 = 25 \Rightarrow a = 21$ $b = \frac{3(21)+137}{21+4} = \frac{63+137}{25} = \frac{200}{25} = 8$ $|21 - 8| = 13$ - **Case 4**: $a + 4 = 125 \Rightarrow a = 121$ $b = \frac{3(121)+137}{121+4} = \frac{363+137}{125} = \frac{500}{125} = 4$ $|121 - 4| = 117$. 5. **Finding the Minimum**: Now we have the values of $|a - b|$ from the valid cases: - From Case 2: 27 - From Case 3: 13 - From Case 4: 117 The minimum value of $|a - b|$ is: $\min(27, 13, 117) = 13$. Thus, the minimal possible value of $|a - b|$ is $\boxed{13}$.

Examples of complete responses from our LLM reflection framework

**Example from GSM8K:**

*Problem:* James creates a media empire. He creates a movie for $2000. Each DVD cost $6 to make. He sells it for 2.5 times that much. He sells 500 movies a day for 5 days a week. How much profit does he make in 20 weeks?

*Incorrect Response:* To solve the problem step-by-step, let's break down the costs, revenues, and profits associated with James's media empire. ### Step 1: Calculate the cost of producing one DVD. The cost to create one DVD is given as $6. ### Step 2: Calculate the selling price of one DVD. James sells each DVD for 2.5 times the cost to make it. We calculate this as follows: Selling Price $= 2.5 \times$ Cost, Selling Price $= 2.5 \times 6 = 15$. ### Step 3: Calculate the profit for one DVD. To find the profit for each DVD sold, we subtract the cost from the selling price: Profit per DVD $=$ Selling Price $-$ Cost, Profit per DVD $= 15 - 6 = 9$. ### Step 4: Calculate the total number of DVDs sold in one week. James sells 500 DVDs a day for 5 days a week. Thus, the total number of DVDs sold in one week is: Total DVDs sold in a week $=$ DVDs sold per day $\times$ Number of days, Total DVDs sold in a week $= 500 \times 5 = 2500$. ### Step 5: Calculate the total profit in one week. Now, we can calculate the total profit for one week by multiplying the profit per DVD by the total number of DVDs sold in a week: Total Profit per week $=$ Profit per DVD $\times$ Total DVDs sold in a week, Total Profit per week $= 9 \times 2500 = 22500$. ### Step 6: Calculate the total profit in 20 weeks. To find the total profit in 20 weeks, we multiply the weekly profit by 20: Total Profit in 20 weeks $=$ Total Profit per week $\times 20$, Total Profit in 20 weeks $= 22500 \times 20 = 450000$. ### Final Answer The total profit James makes in 20 weeks is: $\boxed{450000}$

*Complete Response:* ### Review of Incorrect Explanation: 1. **Initial Setup**: The explanation begins correctly by identifying the cost of producing one DVD as $6, but it misses the initial cost of creating the movie, which is $2000. This cost should be considered in the overall profit calculation. 2. **Profit Calculation**: The profit per DVD calculation is accurate, yielding a profit of $9 per DVD. However, the explanation neglects to take into account the one-time cost of creating the movie when calculating total profit over 20 weeks. 3. **Weekly Sales Calculations**: The calculations for the total number of DVDs sold in one week are accurate, resulting in 2500 DVDs sold. However, the total profit calculation for a week does not account for the initial movie production cost, leading to an inflated total profit over the 20 weeks. 4. **Final Calculation**: The final profit calculation incorrectly suggests that the profit over 20 weeks is simply 22500 multiplied by 20, without factoring in the $2000 initial cost of the movie. ### Corrected Explanation: To solve the problem correctly, we must account for all costs and revenues, including the initial cost associated with creating the movie. #### Step 1: Calculate the cost of producing one DVD. The cost to create one DVD is given as: Cost per DVD $= 6$ dollars. #### Step 2: Calculate the selling price of one DVD. James sells each DVD for 2.5 times its cost: Selling Price $= 2.5 \times$ Cost $= 2.5 \times 6 = 15$ dollars. #### Step 3: Calculate the profit for one DVD. The profit for each DVD sold is calculated as: Profit per DVD $=$ Selling Price $-$ Cost $= 15 - 6 = 9$ dollars. #### Step 4: Calculate the total number of DVDs sold in one week. James sells 500 DVDs a day for 5 days a week: Total DVDs sold in a week $= 500 \times 5 = 2500$. #### Step 5: Calculate the total profit in one week. The total profit for one week can be calculated as: Total Profit per week $=$ Profit per DVD $\times$ Total DVDs sold in a week $= 9 \times 2500 = 22500$ dollars. #### Step 6: Calculate the total profit in 20 weeks. To find the total profit over 20 weeks without considering the initial movie cost: Total Profit in 20 weeks $=$ Total Profit per week $\times 20 = 22500 \times 20 = 450000$ dollars. #### Step 7: Account for the initial cost of creating the movie. Since the initial cost to create the movie is $2000, we need to subtract this from the total profit over 20 weeks: Net Profit $=$ Total Profit in 20 weeks $-$ Initial Cost $= 450000 - 2000 = 448000$ dollars. ### Final Answer: The total profit James makes in 20 weeks, after accounting for the initial cost of creating the movie, is: $\boxed{448000}$

