# OpenReview forum: "PersonaMath: Enhancing Math Reasoning through Persona-Driven Data Augmentation"
_ICLR.cc/2025/Conference — ICLR 2025 Conference Withdrawn Submission_

### Official Review · Reviewer_9pVJ · 2024-10-26

**Soundness:** 2
**Presentation:** 2
**Contribution:** 2
**Rating:** 3
**Confidence:** 4

**Summary:**

This paper presents a novel approach to improving the mathematical reasoning capabilities of open-source language models (LLMs) by leveraging persona-driven data augmentation. Through the PersonaMathQA dataset, the authors aim to enhance model performance on challenging math datasets like MATH and GSM8K using a two-stage augmentation method. The first stage, Persona Diversification, utilizes persona-driven rewrites to increase dataset diversity. The second stage, Reflection, prompts LLMs to learn from previously incorrect answers, enhancing their reasoning capabilities.

**Strengths:**

The model achieves high performance with a smaller dataset than traditional augmentation methods, demonstrating efficient learning from a high-quality, diverse dataset.

The authors will open-source the PersonaMathQA dataset and models, enabling further research and development in the field.

**Weaknesses:**

1. In Table 3, the method seems to be effective only when the base model is weak, like LLaMA-2-7B, when the base model is Qwen2.5-7B, the PersonaMathQA even brings a negative impact on GSM8k.

2. Baselines are insufficient. There are more datasets like Orca-Math, Xwin-Math, DART-Math, etc.

3. Regarding Learn from Reflection, the paper lacks references like [1].

4. I am not sure what is the motivation of applying different personas to solve math problems. Also not clear what is the connection between stage 1 and 2, why we need to connect persona diversification with answer reflection.

5. The paper seems to focus on persona-enhanced math reasoning, but I cannot see many experiments on the persona selection, distribution and impact. It is better to show the readers what kind of different personas can improve math reasoning.

[1] Learn Beyond The Answer: Training Language Models with Reflection for Mathematical Reasoning. Zhang et al.

**Questions:**

N/A

---

### Official Review · Reviewer_wYSc · 2024-10-30

**Soundness:** 2
**Presentation:** 3
**Contribution:** 2
**Rating:** 3
**Confidence:** 4

**Summary:**

This study aims to improve the mathematical reasoning abilities of LLMs by introducing a Persona-Driven Method for data synthesis.

This approach utilizes the GSM8k and MATH datasets (which include golden results) and consists of two main parts:
1. Using Persona Prompts to rewrite questions (as seen in lines 212-215) and requesting the LLMs (GPT4o-mini) to generate Chain of Thought (CoT) solutions.
2. Rewriting the solutions through Persona Prompts (as seen in lines 250-257), based on the mathematical problems and incorrect explanations generated from stage 1.

All synthesized data is presented in Table 2.

The current work utilizes LLaMA2-7B, LLaMA-3.1-8B, and Qwen2.7-7B as backbones to conduct experiments on the GSM8k and MATH datasets. Ablation studies compare the diversity of the synthesized data using Word Types and TTR metrics, as well as the performance gains resulting from different stages.

**Strengths:**

1. The paper is well-written, and the experiments are clear and easy to understand.
2. This approach explores a rewriting strategy for given problems and solutions, and the rewritten data can enhance model performance.
3. This work may release some datasets that could support the community in training new LLMs for math.

**Weaknesses:**

1. The motivation of the paper is worth discussing—specifically, whether persona data is beneficial for mathematical tasks. As is well known, math tasks are objective and logical, while persona data aims to enhance diversity and subjectivity. Moreover, the authors do not provide sufficient evidence to support the idea behind this work.

2. The authors may be inspired by related work [1], which also utilizes persona-driven methods for data synthesis. However, it is important to note that the primary aim of method [1] is scaling, while the current work focuses on rewriting data. This distinction means that the current method does not fully leverage the advantages of persona-driven methods.

3. The authors claim to achieve state-of-the-art (SOTA) results on the MATH dataset, but they may have overlooked some relevant works on data synthesis [2] [3] [4].

4. The officially reported result for LLaMA-3.1-8B is 51.9 (zero-shot CoT), which appears to indicate a performance decline compared to the fine-tuning results presented in the paper.

[1] Scaling synthetic data creation with 1,000,000,000 personas

[2] DART-Math: Difficulty-Aware Rejection Tuning for Mathematical Problem-Solving

[3] JiuZhang3.0: Efficiently Improving Mathematical Reasoning by Training Small Data Synthesis Models

[4] OpenMathInstruct-2: Accelerating AI for Math with Massive Open-Source Instruction Data

**Questions:**

Why is Word Types considered an effective evaluation metric? Shouldn't mathematical problems take into account the diversity of the problem-solving process or thought process, rather than just the diversity of expression?

---

### Official Review · Reviewer_Ah8F · 2024-10-30

**Soundness:** 2
**Presentation:** 2
**Contribution:** 2
**Rating:** 5
**Confidence:** 4

**Summary:**

This paper introduces a novel data augmentation method, called PersonaMath, to improvelarge language models (LLMs) in mathematical problem-solving. PersonaMath is a two-stage persona-driven approach, stage one employs a closed-source LLM to generate CoT, and then rewrites questions using diverse personas to boost question variety. Stage two applies a reflection mechanism for the LLM to learn from incorrect answers, increasing model performance on challenging questions.
The created data is called PersonaMathQA, which includes 70.3K data examples. It enables PersonaMath-7B (based on LLaMA-2-7B) to achieve superior accuracy on MATH and GSM8K benchmarks.

**Strengths:**

1. Persona-driven augmentation creates a smaller but highly effective dataset, leading to better model performance at lower computational costs.

2. The PersonaMath-7B model achieves higher accuracy on math benchmarks (GMS8K and MATH).

**Weaknesses:**

1. The reflection part closely resembles the approach in [1], the novelty of this paper is not clearly differentiated from pervious work.

[1] Learn Beyond The Answer: Training Language Models with Reflection for Mathematical Reasoning. EMNLP 2024.


2. As demonstrated in paper [1], various augmentation methods can achieve approximately 70% accuracy on 7B models (prior to LLaMA-3). It is unclear why the performance here is lower than those methods.

3. How does the persona generation strategy in this paper differ from that in paper [2]?

[2] Scaling Synthetic Data Creation with 1,000,000,000 Personas

**Questions:**

1. Why not compare with RefAug [1]? What is the novel contribution in reflection part compared with [1]?

[1] Learn Beyond The Answer: Training Language Models with Reflection for Mathematical Reasoning. EMNLP 2024.

2. Why the performance here is lower than those methods in [1]?

---

### Official Review · Reviewer_fgho · 2024-11-03

**Soundness:** 2
**Presentation:** 2
**Contribution:** 1
**Rating:** 3
**Confidence:** 4

**Summary:**

To improve large language models' mathematical reasoning ability, the authors introduce PersonaMathQA, a dataset derived from MATH and GSM8K, using a two-stage approach, the first one is persona diversification which generates detailed solutions and enhances dataset diversity, and the other is reflection which utilizes more challenging questions for better learning.
They also proposed PersonaMath-7B model, based on LLaMA-2-7B, which achieves 24.2% accuracy on MATH and 68.7% on GSM8K, surpasses baselines, and achieves SOTA despite using fewer data points. The dataset and models are open-sourced for public use.

**Strengths:**

The paper introduces a new method for augmenting mathematical reasoning dataset and achieves enhanced performance through fine-tuning on several benchmarks.

**Weaknesses:**

Augmenting datasets based on persona is not a novel technique and is already proved effective by previous research, as the authors cite in the paper: XinChan, Xiaoyang Wang, Dian Yu, Haitao Mi,and Dong Yu. Scaling synthetic data creation with 1,000,000,000 personas, 2024.
I don't see any contribution from this paper, maybe the reflection part? Even though yes, reflection or self-revising is a well-established method in enhancing the reasoning ability of LLMs from inference sides. The authors fail to demonstrate the novel insight of including this method, therefore, it seems a direct implementation from previous work.

**Questions:**

I really hope the author can demonstrate where this work distinguishes from previous research. I would be helpful in case of my misunderstanding.

---

### Official Review · Reviewer_jnpR · 2024-11-04

**Soundness:** 3
**Presentation:** 3
**Contribution:** 2
**Rating:** 5
**Confidence:** 4

**Summary:**

This paper presents PersonaMathQA, a new dataset designed to improve math skills in open-source language models. This paper introduces two approaches to data augmentation—diverse persona-based question rewriting and reflection-based solution correction. The PersonaMath model achieves top performance on math benchmarks with less data than other datasets. The dataset, model, and code are released for public use.

**Strengths:**

1. This paper is well written and easy to follow.
2. The experiments show good performance on different base models.
3. The open resources support further research within the community.

**Weaknesses:**

1. Lack of innovation: The paper shows minimal methodological innovation compared to previous data augmentation work, primarily combining existing approaches for math problem augmentation rather than introducing new techniques. It directly uses data from Persona Hub for question rewriting, and a detailed discussion on math problem synthesis already provided in the Persona Hub paper—Scaling Synthetic Data Creation with 1,000,000,000 Personas (https://arxiv.org/abs/2406.20094).
2. Limited detail on persona-driven augmentation: The discussion on the use of Persona data is insufficient. There is no clear explanation of how the 200k Persona data points were selected for augmentation.
3. Limited impact of reflection method: The reflection method only corrected 10% of incorrect problems in stage 2. While performance on GSM8K improved by 2.5% (as shown in Figure 4), performance on MATH slightly declined, which contradicts the claim that reflection significantly helps with more challenging problems.
4. Insufficient experiments: Reporting results on additional math datasets beyond MATH and GSM8K, and comparing with more data augmentation methods, could strengthen the paper’s persuasiveness.

**Questions:**

Please refer to the above Weaknesses section.

---

### Note · Authors · 2024-11-19

I have read and agree with the venue's withdrawal policy on behalf of myself and my co-authors.